# Structured Prediction for Conditional Meta-Learning

**Ruohan Wang,   Yiannis Demiris,   Carlo Ciliberto**
Dept. of Electrical and Electronic Engineering
Imperial College London
London, UK
{r.wang16,y.demiris,c.ciliberto}@imperial.ac.uk

## Abstract

The goal of optimization-based meta-learning is to find a single initialization shared across a distribution of tasks to speed up the process of learning new tasks. *Conditional* meta-learning seeks task-specific initialization to better capture complex task distributions and improve performance. However, many existing conditional methods are difficult to generalize and lack theoretical guarantees. In this work, we propose a new perspective on conditional meta-learning via structured prediction. We derive *task-adaptive structured meta-learning* (TASML), a principled framework that yields task-specific objective functions by weighing meta-training data on target tasks. Our non-parametric approach is model-agnostic and can be combined with existing meta-learning methods to achieve conditioning. Empirically, we show that TASML improves the performance of existing meta-learning models, and outperforms the state-of-the-art on benchmark datasets.

## 1   Introduction

State-of-the-art learning algorithms such as neural networks typically require vast amounts of data to generalize well. This is problematic for applications with limited data availability (e.g. drug discovery [3]). Meta-learning is often employed to tackle the lack of training data [17, 40, 53]. It is designed to learn data-driven inductive bias to speed up learning on novel tasks [50, 52], with many application settings such as learning-to-optimize [30], few-shot learning [16, 27] and hyperparameter optimization [18]. Meta-learning methods could be broadly categorized into metric-learning [e.g. 37, 47, 53], model-based [e.g. 20, 22, 30], and optimization-based [e.g. 17, 35, 44].

We focus on optimization-based approaches, which cast meta-learning as a bi-level optimization [4, 17, 39]. At the single-task level, an "inner" algorithm performs task-specific optimization starting from a set of meta-parameters shared across all tasks. At the "outer" level, a meta learner accrues experiences from observed tasks to learn the aforementioned meta-parameters. These methods seek to learn a single initialization of meta-parameters that can be effectively adapted to all tasks. Relying on the shared initialization is challenging for complex (e.g. multi-modal) task distributions, since different tasks may require a substantially different initialization, given the same adaptation routine. This makes it infeasible to find a common meta-parameters for all tasks. Several recent works [10, 15, 23, 24, 28, 44, 54, 55, 56] address the issue by conditioning such parameters on target tasks, and demonstrate consistent improvements over unconditional meta-learning. However, existing methods often lack theoretical guarantees on generalization performance and implements specific conditioning principles with customized networks, which may be difficult to generalize across different application settings.

In this paper, we offer a novel perspective on *conditional* meta-learning based on structured prediction [6]. This enables us to propose *Task-adaptive Structured Meta-Learning (TASML)* – a general framework for conditional meta-learning – by interpreting the inner algorithm as the structured output to be predicted, conditioned on target tasks. We derive a principled estimator that minimizes

task-specific meta-learning objectives, which weigh known training tasks based on their similarities with the target task. The proposed framework is non-parametric and thus requires access to training data at test time for the task-specific objectives. We introduce an efficient algorithm for TASML to mitigate the additional computational costs associated with optimizing these task-specific objectives. Intuitively, the proposed framework learns a target task by explicitly recalling only the most relevant tasks from past observations, to better capture the local task distribution for improved generalization. The relevance of previously observed tasks with respect to the target one is measured by a structured prediction approach from [13]. TASML is model-agnostic and can easily adapt existing meta-learning methods to achieve conditional meta-learning.

We empirically evaluate TASML on several competitive few-shot classification benchmarks, including datasets derived from IMAGENET and CIFAR respectively. We show that TASML outperforms state-of-the-art methods, and improves the accuracy of existing meta-learning algorithms by adapting them into their respective conditional variants. We also investigate TASML's trade-off between computational efficiency and accuracy improvement, showing that the proposed method achieves good efficiency in learning new tasks.

Our main contributions include: $i$) a new perspective on conditional meta-learning based on structured prediction, $ii$) TASML, a conditional meta-learning framework that generalizes existing meta-learning methods, $iii$) a practical and efficient algorithm under the proposed framework, and $iv$) a thorough evaluation of TASML on benchmarks, outperforming state-of-the-art methods.

## 2 Background and Notation

For clarity, in the following we focus on meta-learning for supervised learning. However, the discussion below and our proposed approach also apply to general learning settings.

**Supervised learning.** In supervised learning, given a probability distribution $\rho$ over two spaces $\mathcal{X} \times \mathcal{Y}$ and a loss $\ell : \mathcal{Y} \times \mathcal{Y} \to \mathbb{R}$ measuring prediction errors, the goal is to find $f : \mathcal{X} \to \mathcal{Y}$ that minimizes the *expected risk*

$$\min_{f:\mathcal{X} \to \mathcal{Y}} \mathcal{E}(f) \quad \text{with} \quad \mathcal{E}(f) = \mathbb{E}_\rho \, \ell(f(x), y), \tag{1}$$

with $(x, y)$ sampled from $\rho$. A finite training set $D = (x_j, y_j)_{j=1}^m$ of *i.i.d* samples from $\rho$ is given. A learning algorithm typically finds $f \in \mathcal{F}$ within a prescribed set $\mathcal{F}$ of candidate models (e.g. neural networks, reproducing kernel Hilbert spaces), by performing empirical risk minimization on $D$ or adopting online strategies such as stochastic gradient methods (SGD) (see [46] for an in-depth view on statistical learning). A learning algorithm may thus be seen as a function $\text{Alg} : \mathcal{D} \to \mathcal{F}$ that maps an input dataset $D$ to a model $f = \text{Alg}(D)$, where $\mathcal{D}$ is the space of datasets on $\mathcal{X} \times \mathcal{Y}$.

**Meta-learning.** While in supervised settings $\text{Alg}(\cdot)$ is chosen a-priori, the goal of meta-learning is to *learn a learning algorithm* suitable for a family of tasks. Thus, we consider a parametrization $\text{Alg}(\theta, \cdot) : \mathcal{D} \to \mathcal{F}$ for the inner algorithm, with $\theta \in \Theta$ a space of meta-parameters and aim to solve

$$\min_{\theta \in \Theta} \mathcal{E}(\theta) \quad \text{with} \quad \mathcal{E}(\theta) = \mathbb{E}_\mu \mathbb{E}_\rho \, \mathcal{L}\big(\text{Alg}(\theta, D^{tr}), D^{val}\big). \tag{2}$$

Here, $\rho$ is a task distribution sampled from a meta-distribution $\mu$, and $D^{tr}$ and $D^{val}$ are respectively training and validation sets of *i.i.d* data points $(x, y)$ sampled from $\rho$. The task loss $\mathcal{L} : \mathcal{F} \times \mathcal{D} \to \mathbb{R}$ is usually an empirical average of the prediction errors on a dataset according to an inner loss $\ell$,

$$\mathcal{L}(f, D) = \frac{1}{|D|} \sum_{(x,y) \in D} \ell(f(x), y), \tag{3}$$

with $|D|$ the cardinality of $D$. We seek the best $\theta^*$ such that applying $\text{Alg}(\theta^*, \cdot)$ on $D^{tr}$ achieves lowest generalization error on $D^{val}$, among all algorithms parametrized by $\theta \in \Theta$. In practice, we have access to only a finite meta-training set $S = (D_i^{tr}, D_i^{val})_{i=1}^N$ and the meta-parameters $\hat{\theta}$ are often learned by (approximately) minimizing

$$\hat{\theta} = \underset{\theta \in \Theta}{\text{argmin}} \; \frac{1}{N} \sum_{i=1}^N \mathcal{L}\big(\text{Alg}(\theta, D_i^{tr}), D_i^{val}\big). \tag{4}$$

Meta-learning methods address (4) via first-order methods such as SGD, which requires access to $\nabla_\theta \text{Alg}(\theta, D)$, the (sub)gradient of the inner algorithm over its meta-parameters. For example, model-agnostic meta-learning (MAML) [17] and several related methods [e.g. 4, 31, 39] cast meta-learning as a bi-level optimization problem. In MAML, $\theta$ parametrizes a model $f_\theta : \mathcal{X} \to \mathcal{Y}$ (e.g. a neural network), and $\text{Alg}(\theta, D)$ performs one (or more) steps of gradient descent minimizing the empirical risk of $f_\theta$ on $D$. Formally, given a step-size $\eta > 0$,

$$f_{\theta'} = \text{Alg}(\theta, D) \quad \text{with} \quad \theta' = \theta - \eta \, \nabla_\theta \mathcal{L}(f_\theta, D).$$

Inspired by MAML, meta-representation learning [9, 58] performs task adaptation via gradient descent on only a subset of model parameters and considers the remaining ones as a shared representation.

## 3 Conditional Meta-Learning

Although remarkably efficient in practice, optimization-based meta-learning typically seeks a single set of meta-parameters $\theta$ for all tasks from $\mu$. This shared initialization might be limiting for complex (e.g. multi-modal) task distributions: dissimilar tasks require substantially different initial parameters given the same task adaptation routine, making it infeasible to find a common initialization [54, 56]. To address this issue, several recent works learn to condition the initial parameters on target tasks (see Fig. 1 in Sec. 4 for a pictorial illustration of this idea). For instance, [44, 54, 56] directly learn data-driven mappings from target tasks to initial parameters, and [24] conditionally transforms feature representations based on a metric space trained to capture inter-class dependencies. Alternatively, [23] considers a mixture of hierarchical Bayesian models over the parameters of meta-learning models to condition on target tasks, while [10] preliminarily explores task-specific initialization by optimizing weighted objective functions. However, these existing methods typically implement specific conditional principles with customized network designs, which may be difficult to generalize to different application settings. Further, they often lack theoretical guarantees.

**Conditional Meta-learning.** We formalize the conditional approaches described above as *conditional meta-learning*. Specifically, we condition the meta-parameters $\theta$ on $D$ by parameterizing $\text{Alg}(\tau(D), \cdot)$ with $\tau(D) \in \Theta$, a meta-parameter valued function. We cast *conditional meta-learning* as a generalization of (2) to minimize

$$\min_{\tau : \mathcal{D} \to \Theta} \mathcal{E}(\tau) \quad \text{with} \quad \mathcal{E}(\tau) = \mathbb{E}_\mu \mathbb{E}_\rho \, \mathcal{L}\Big( \text{Alg}\big( \tau(D^{tr}), \, D^{tr} \big), \, D^{val} \Big), \tag{5}$$

over a suitable space of functions $\tau : \mathcal{D} \to \Theta$ mapping datasets $D$ to algorithms $\text{Alg}(\tau(D), \cdot)$. While (5) uses $D^{tr}$ for both the conditioning and inner algorithm, more broadly $\tau$ may depend on a separate dataset $\tau(D^{con})$ of "contextual" information (as recently investigated also in [15]), similar to settings like collaborative filtering with side-information [1]. Note that standard (unconditional) meta-learning can be interpreted as an instance of (5) with $\tau(D) \equiv \theta$, the constant function associating every dataset $D$ to the same meta-parameters $\theta$. Intuitively, we can expect a significant improvement from the solution $\tau_*$ of (5) compared to the solution $\theta_*$ of (2), since by construction $\mathcal{E}(\tau_*) \leq \mathcal{E}(\theta_*)$.

Conditional meta-learning leverages a finite number of meta-training task to learn $\tau : \mathcal{D} \to \Theta$. While it is possible to address this problem in a standard supervised setting, we stress that meta-learning poses unique challenges from both modeling and computational perspectives. A critical difference is the output set: in standard settings, this is usually a linear space (namely $\mathcal{Y} = \mathbb{R}^k$), for which there exist several methods to parameterize suitable spaces of hypotheses $f : \mathcal{X} \to \mathbb{R}^k$. In contrast, when the output space $\Theta$ is a complicated, "structured" set (e.g. space of deep learning architectures), it is less clear how to find a space of hypotheses $\tau : \mathcal{D} \to \Theta$ and how to perform optimization over them. These settings however are precisely what the literature of *structured prediction* aims to address.

### 3.1 Structured Prediction for Meta-learning

Structured prediction methods are designed for learning problems where the output set is not a linear space but rather a set of structured objects such as strings, rankings, graphs, 3D structures [6, 36]. For conditional meta-learning, the output space is a set of inner algorithms parameterized by $\theta \in \Theta$. Directly modeling $\tau : \mathcal{D} \to \Theta$ can be challenging. A well-established strategy in structured prediction is therefore to first learn a joint function $T : \Theta \times \mathcal{D} \to \mathbb{R}$ that, in our setting, measures the quality of a model $\theta$ for a specific dataset $D$. The structured prediction estimator $\tau$ is thus defined as

the function choosing the optimal model parameters $\tau(D) \in \Theta$ given the input dataset $D$

$$\tau(D) = \operatorname*{argmin}_{\theta \in \Theta} \; T(\theta, D). \tag{6}$$

Within the structured prediction literature, several strategies have been proposed to model and learn the joint function $T$, such as SVMStruct [51] and Maximum Margin Markov Networks [49]. However, most methods have been designed to deal with output spaces $\Theta$ that are discrete or finite and are therefore not suited for conditional meta-learning. To our knowledge, the only structured prediction framework capable of dealing with general output spaces (e.g. dense set $\Theta$ of network parameters) is the recent work based on the *structure encoding loss function* principle [13, 14]. This approach also enjoys strong theoretical guarantees including consistency and learning rates. We propose to apply such a method to conditional meta-learning and then study its generalization properties.

**Task-adaptive Structured Meta-Learning.** To apply [13], we assume access to a reproducing kernel [5] $k : \mathcal{D} \times \mathcal{D} \to \mathbb{R}$ on the space of datasets (see (10) in Sec. 4.1 for an example). Given a meta-training set $S = (D_i^{tr}, D_i^{val})_{i=1}^N$ and a new task $D$, the structured prediction estimator is

$$\tau(D) = \operatorname*{argmin}_{\theta \in \Theta} \; \sum_{i=1}^N \alpha_i(D) \, \mathcal{L}\big(\mathrm{Alg}(\theta, D_i^{tr}), \; D_i^{val}\big) \tag{7}$$

$$\text{with} \quad \alpha(D) = (\mathbf{K} + \lambda I)^{-1} v(D) \in \mathbb{R}^N,$$

where $\lambda > 0$ is a regularizer, $\alpha_i(D)$ denotes the $i$-th entry of the vector $\alpha(D)$ while $\mathbf{K} \in \mathbb{R}^{N \times N}$ and $v(D) \in \mathbb{R}^N$ are the kernel matrix and evaluation vector with entries $\mathbf{K}_{i,j} = k(D_i^{tr}, D_j^{tr})$ and $v(D)_i = k(D_i^{tr}, D)$, respectively. We note that (7) is an instance of (6), where the joint functional $T$ is modelled according to [13] and learned on the meta-training set $S$. The resulting approach is a non-parametric method for conditional meta-learning, which accesses training tasks for solving (7).

We refer to the estimator in (7) as *Task-adaptive Structured Meta-Learning (TASML)*. In this formulation, we seek $\theta$ to minimize a weighted meta-learning objective, where the $\alpha : \mathcal{D} \to \mathbb{R}^N$ can be interpreted as a "scoring" function that measures the relevance of known training tasks to target tasks. The structured prediction process is hence divided into two distinct phases: $i$) a *learning* phase for estimating the scoring function $\alpha$ and $ii$) a *prediction* phase to obtain $\tau(D)$ by solving (7) on $D$. The following remark draws a connection between TASML and unconditional meta-learning methods.

**Remark 1** (Connection with standard meta-learning). *The objective in (7) recovers the empirical risk minimization for meta-learning introduced in (4) if we set $\alpha_i(D) \equiv 1$. Hence, methods from Sec. 2 – such as MAML – can be interpreted as conditional meta-learning algorithms that assume all tasks being equally related to one other.*

Remark 1 suggests that TASML is compatible with most existing (unconditional) meta-learning methods in the form of (4) (or its stochastic variants). Thus, TASML can leverage existing algorithms, including their architectures and optimization routines, to solve the weighted meta-learning problem in (7). In Sec. 5, we show empirically that TASML improves the generalization performance of three meta-learning algorithms by adopting the proposed structured prediction perspective. Additionally, in Sec. 5.5 we discuss the potential relations between (7) and recent multi-task learning (MTL) strategies that rely on task-weighing [12, 25].

**Theoretical Properties.** Thanks to adopting a structured prediction perspective, we can characterize TASML's learning properties. In particluar, the following result provides non-asymptotic excess risk bounds for our estimator that indicate how fast we can expect the prediction error of $\tau$ to decrease as the number $N$ of meta-training tasks grows.

**Theorem 1** (Informal – Learning Rates for TASML). *Let $S = (D_i^{tr}, D_i^{val})_{i=1}^N$ be sampled from a meta-distribution $\mu$ and $\tau_N$ the estimator in (7) trained with $\lambda = N^{-1/2}$ on $S$. Then, with high probability with respect to $\mu$,*

$$\mathcal{E}(\tau_N) \; - \inf_{\tau : \mathcal{D} \to \Theta} \mathcal{E}(\tau) \; \leq O(N^{-1/4}). \tag{8}$$

Thm. 1 shows that the proposed algorithm asymptotically yields the *best* possible task-conditional estimator for the family of tasks identified by $\mu$, over the novel samples from validation sets. The proof of Thm. 1 leverages recent results from the literature on structured prediction [32, 43], combined with standard regularity assumptions on the meta-distribution $\mu$. See Appendix A for a proof and further discussion on the relation between TASML and general structured prediction.

# 4 A Practical Algorithm for TASML

The proposed TASML estimator $\tau : \mathcal{D} \to \Theta$ offers a principled approach to conditional meta-learning. However, task-specific objectives incur additional computational cost compared to unconditional meta-learning models like MAML, since we have to repeatedly solve (7) for each target task $D$, in particular when the number $N$ of meta-training tasks is large. We hence introduce several adjustments to TASML that yield a significant speed-up in practice, without sacrificing overall accuracy.

**Initialization by Meta-Learning.** Following the observation in Remark 1, we propose to learn an "agnostic" $\hat{\theta} \in \Theta$ as model initialization before applying TASML. Specifically, we obtain $\hat{\theta}$ by applying a standard (unconditional) meta-learning method solving (4). We then initialize the inner algorithm with $\hat{\theta}$, followed by minimizing (7) over the meta-parameters. In practice, the learned initialization significantly speeds up the convergence in (7). We stress that the proposed initialization is optional: directly applying TASML to each task using random initialization obtains similar performance, although it takes more training iterations to achieve convergence (see Appendix C.1).

**Top-$M$ Filtering.** In (7), the weights $\alpha_i(D)$ measure the relevance of the $i$-th meta-training task $D_i^{tr}$ to the target task $D$. We propose to keep only the top-$M$ values from $\alpha(D)$, with $M$ a hyperparameter, and set the others to zero. This filtering reduces the computational cost of (7), by constraining training to only tasks $D_i^{tr}$ most relevant to $D$, The filtering process has little impacts on the final performance, since we empirically observed that only a small percentage of tasks have large $\alpha(D)$, given a large number $N$ of training tasks (e.g. $N > 10^5$). In our experiments we chose $M$ to be $\sim 1\%$ of $N$, since larger values did not provide significant improvements in accuracy (see Appendix C.2 for further ablation). We observe that TASML's explicit dependence on meta-training tasks resembles meta-learning methods with external memory module [45], The proposed filtering process in turn resembles memory access rules but requires no learning. For each target task, only a small number of training tasks are accessed for adaptation, limiting the overall memory requirements.

**Task Adaptation.** The output $\tau(D)$ in (7) depends on the target task $D$ only via the task weights $\alpha(D)$. We propose to directly optimize $D$ with an additional term $\mathcal{L}(\mathrm{Alg}(\theta, D), D)$, which encourages $\theta$ to directly exploit the training signals in target task $D$ and achieve small empirical error on it. The extra term may be interpreted as adding a "special" training task $(\tilde{D}^{tr}, \tilde{D}^{val}) = (D, D)$, in which the support set and query set coincides, to the meta-training set $(D_i^{tr}, D_i^{val})_{i=1}^N$

$$\tau(D) \;=\; \underset{\theta \in \Theta}{\mathrm{argmin}} \; \sum_{i=1}^N \alpha_i(D) \, \mathcal{L}\big(\mathrm{Alg}(\theta, D_i^{tr}), \, D_i^{val}\big) \;+\; \mathcal{L}\big(\mathrm{Alg}(\theta, D), \, D\big), \qquad (9)$$

By construction, the additional task offers useful training signals by regularizing the model to focus on relevant features shared among the target task and past experiences. We refer to Appendix C.3 for an ablation study on how (9) affects test performance.

Alg. 1 implements TASML with the proposed improvements. We initialize the model with meta-parameters $\theta$, by solving the (unconditional) meta-learning problem in (4) over a meta-training set $S = (D_i^{tr}, D_i^{val})_{i=1}^N$. We also learn the scoring function $\alpha$ according to (7) by inverting the kernel matrix $\mathbf{K}$. While this is an expensive step of up to $O(N^3)$ in complexity, sketching methods may be used to significantly speed up the inversion without loss of accuracy [33, 42]. We stress that the scoring function $\alpha$ is learned only once for all target tasks. For any target task $D$, we compute $\alpha(D)$ and only keep the top-$M$ tasks $S_M \subset S$ with largest $\alpha_i(D)$. Lastly, we minimize (9) over $S_M$.

## 4.1 Implementation Details

**Reproducing Kernel on Datasets.** TASML requires a positive definite kernel $k : \mathcal{D} \times \mathcal{D} \to \mathbb{R}$ to learn the score function $\alpha : \mathcal{D} \to \mathbb{R}^N$ in (7). In this work, we take $k$ as the Gaussian kernel of the *maximum mean discrepancy (MMD)* [19] of two datasets. MMD is a popular distance metric on datasets or distributions. More precisely, given two datasets $D, D' \in \mathcal{D}$ and a feature map $\varphi : \mathcal{X} \to \mathbb{R}^p$ on input data, we consider the kernel $k : \mathcal{D} \times \mathcal{D} \to \mathbb{R}$ defined as

$$k(D, D') = \exp\Big( - \|\bar{\varphi}(D) - \bar{\varphi}(D')\|^2 / \sigma^2 \Big) \qquad \text{with} \qquad \bar{\varphi}(D) = \frac{1}{|D|} \sum_{(x,y) \in D} \varphi(x) \qquad (10)$$

where $\sigma > 0$ is a bandwidth parameter and $\bar{\varphi}(D)$ is the *kernel mean embedding* (or signature) of a dataset $D$ with respect to $\varphi$.

**Input:** meta-train set $S = (D_i^{tr}, D_i^{val})_{i=1}^{N}$, initial parameters $\theta$, Filter size $M$, step-size $\eta$, kernel $k$, regularizer $\lambda$

**Meta-Train**:
  Compute the kernel matrix $\mathbf{K} \in \mathbb{R}^{N \times N}$ on $S$
  Let $v : \mathcal{D} \rightarrow \mathbb{R}^N$ where $v(\cdot)_i = k(D_i^{tr}, \cdot)$
  Let $\alpha(\cdot) = (\mathbf{K} + \lambda I)^{-1} v(\cdot)$.
  Initialize $\theta$ with (4) (Optional)

**Meta-Test** with target task $D$:
  Compute task weights $\alpha(D) \in \mathbb{R}^N$.
  Get top-$M$ tasks $S_M \subset S$ with highest $\alpha(D)$

  **while** not converged **do**
    Sample a mini-batch $S_B \subset S_M$ i.i.d.
    Compute gradient $\nabla_\theta$ of (9) over $S_B$.
    $\theta \leftarrow \theta - \eta \, \nabla_\theta$
  **end while**

**Return** $\theta$

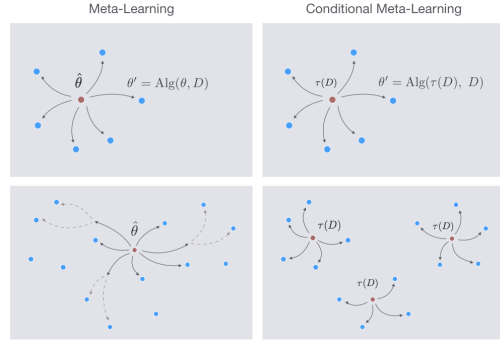

Figure 1: Comparing unconditioned (left) and conditional (right) meta-learning. Unconditional methods using a shared initialization (red dot) may fail to adapt to dissimilar tasks (outer blue dots, bottom left). Conditional meta-learning can handle such setting via adaptive initialization (red dots, bottom right).

The map $\varphi$ plays a central role. It can be either fixed a priori or learned, depending on the application. A good practice when working with MMD is to use a characteristic kernel [48]. In our experiments, the Gaussian kernel in (10) led to best performance (see Appendix C.4 for other options). We note that, more generally, $\varphi$ could also be taken to be a joint feature map on both input and output. We describe our choice of feature map $\varphi$ below.

**Pre-trained Feature Map.** Expressive input representation plays a significant role in meta-learning, and may be obtained via standard supervised learning (see e.g. [41, 44]). We choose the pre-trained feature map from [44] for the kernel mean embedding $\bar{\varphi}(D)$ in (10) and input representation $\varphi(x)$.

**Model Architecture.** As observed in Remark 1, TASML is compatible with a wide range of existing meta-learning algorithms. In Sec. 5.3, we report on several implementations of Alg. 1, leveraging architectures and optimization techniques from existing meta-learning methods. In addition, we introduce LS META-LEARN, a least-squares meta-learning algorithm that is highly effective in combination with TASML. Similar to [9], we choose an "inner" algorithm that solves a least-squares objective $\ell(y, y') = \|y - y'\|^2$ in closed-form. We propose to induce the *task loss* $\mathcal{L}$ in (3) by the same least-squares objective. We note that while least-squares minimization is not a standard approach in classification settings, it is theoretically principled (see e.g. [7, 34]) and provides a significant improvement to classification accuracy. See Appendix B.1 for more details.

## 5 Experiments

We perform experiments[1] on *C*-way-*K*-shot learning within the episodic formulation of [53]. In this setting, train-validation pairs $(D^{tr}, D^{val})$ are sampled as described in Sec. 2. $D^{tr}$ is a $C$-class classification problem with $K$ examples per class. $D^{val}$ contains samples from the same $C$ classes for estimating model generalization and training meta-learner. We evaluate the proposed method against a wide range of meta-learning algorithms on three few-shot learning benchmarks: the *mini*IMAGENET, *tiered*IMAGENET and CIFAR-FS datasets. We consider the commonly used 5-way-1-shot and 5-way-5-shot settings. For training, validation and testing, we sample three separate meta-datasets $S^{tr}, S^{val}$ and $S^{ts}$, each accessing a disjoint set of classes (e.g. no class in $S^{ts}$ appears in $S^{tr}$ or $S^{val}$). To ensure fair comparison, we adopt the same training and evaluation setup as [44]. Appendix B reports further experimental details including network specification and hyperparameter choice.

Table 1: Classification Accuracy of meta-learning models on *mini*IMAGENET and *tiered*IMAGENET.

| | ACCURACY (%) | | | |
| | *mini*IMAGENET | | *tiered*IMAGENET | |
| UNCONDITIONAL METHODS | 1-SHOT | 5-SHOT | 1-SHOT | 5-SHOT |
| --- | --- | --- | --- | --- |
| MAML [17] | $48.70 \pm 1.84$ | $63.11 \pm 0.92$ | $51.67 \pm 1.81$ | $70.30 \pm 0.8$ |
| IMAML [39] | $49.30 \pm 1.88$ | - | - | - |
| REPTILE [35] | $49.97 \pm 0.32$ | $65.99 \pm 0.58$ | - | - |
| R2D2 [9] | $51.90 \pm 0.20$ | $68.70 \pm 0.20$ | - | - |
| CAVIA [58] | $51.82 \pm 0.65$ | $65.85 \pm 0.55$ | - | - |
| (QIAO ET AL.) [38] | $59.60 \pm 0.41$ | $73.74 \pm 0.19$ | - | - |
| META-SGD [31](LEO FEAT.) | $54.24 \pm 0.03$ | $70.86 \pm 0.04$ | $62.95 \pm 0.03$ | $79.34 \pm 0.06$ |
| CONDITIONAL METHODS | | | | |
| (JERFEL ET AL.) [23] | $51.46 \pm 1.68$ | $65.00 \pm 0.96$ | - | - |
| HSML [56] | $50.38 \pm 1.85$ | - | - | - |
| MMAML [54] | $46.1 \pm 1.63$ | $59.8 \pm 1.82$ | - | - |
| CAML [24] | $59.23 \pm 0.99$ | $72.35 \pm 0.71$ | - | - |
| LEO [44] | $61.76 \pm 0.08$ | $77.59 \pm 0.12$ | $66.33 \pm 0.05$ | $81.44 \pm 0.09$ |
| LEO (LOCAL) [44] | $60.37 \pm 0.74$ | $75.36 \pm 0.44$ | $65.11 \pm 0.72$ | $79.70 \pm 0.59$ |
| TASML (OURS) | $\mathbf{62.04 \pm 0.52}$ | $\mathbf{78.22 \pm 0.47}$ | $\mathbf{66.42 \pm 0.37}$ | $\mathbf{82.62 \pm 0.31}$ |

Table 2: Classification Accuracy of meta-learning models on CIFAR-FS.

| | 1-SHOT | 5-SHOT |
| --- | --- | --- |
| MAML [17] | $58.9 \pm 1.9$ | $71.5 \pm 1.0$ |
| R2D2 [9] | $65.3 \pm 0.2$ | $79.4 \pm 0.1$ |
| PROTONET(RESNET12 FEAT.) [47] | $72.2 \pm 0.7$ | $83.5 \pm 0.5$ |
| METAOPTNET [29] | $72.0 \pm 0.7$ | $84.2 \pm 0.5$ |
| TASML | $\mathbf{74.6 \pm 0.7}$ | $\mathbf{85.1 \pm 0.4}$ |

## 5.1 Experiments on IMAGENET derivatives

We compared TASML with a representative set of baselines on classification accuracy. Unconditional methods include MAML [17], IMAML [39], REPTILE [35], R2D2 [9], (QIAO ET AL. 2018) [38], CAVIA [58], and META-SGD [31] (using [44]'s features). Conditional methods include (JERFELET AL 2019) [23], HSML [57], MMAML [54], CAML [24] and LEO [44].

We include results from our local run of LEO using the official implementation. In our experiments, we observed that LEO appeared sensitive to hyperparameter choices, and obtaining the original performance in [44] was beyond our computational budget. For MMAML, we used the official implementation, since [54] did not report performance on *mini*IMAGENET. We did not compare with $\alpha$-MAML [10] since we did not find ImageNet results nor an official implementation. Other results are cited directly from their respective papers.

Tab. 1 reports the mean accuracy and standard deviation of all the methods over 50 runs, with each run containing 200 random test tasks. TASML outperforms the baselines in three out of the four settings, and achieves performance comparable to LEO in the remaining setting. We highlight the comparison between TASML and LEO (local), as they share the identical experiment setups. The identical setups make it easy to attribute any relative performance gains to the proposed framework. We observe that TASML outperforms LEO (local) in all four settings, averaging over 2% improvements in classification accuracy. The results suggest the efficacy of the proposed method.

## 5.2 Experiments on CIFAR-FS

The recently proposed CIFAR-FS dataset [9] is a new few-shot learning benchmark, consisting of all 100 classes from CIFAR-100 [26]. The classes are randomly divided into 64, 16 and 20 for meta-train, meta-validation, and meta-test respectively. Each class includes 600 images of size $32 \times 32$.

Table 3: Effects of structured prediction on *mini*IMAGENET and *tiered*IMAGENET benchmarks. Structured prediction (SP) improves the underlying meta-learning algorithms in all cases.

| | ACCURACY (%) | | | |
| | *mini*IMAGENET | | *tiered*IMAGENET | |
| | 1-SHOT | 5-SHOT | 1-SHOT | 5-SHOT |
|---|---|---|---|---|
| MAML (LEO FEAT.) | $54.12 \pm 1.84$ | $67.58 \pm 0.92$ | $51.28 \pm 1.81$ | $69.80 \pm 0.84$ |
| SP+MAML (LEO FEAT.) | $\mathbf{58.46 \pm 1.56}$ | $\mathbf{74.51 \pm 0.75}$ | $\mathbf{60.89 \pm 1.64}$ | $\mathbf{78.42 \pm 0.73}$ |
| LEO (LOCAL) | $60.37 \pm 0.74$ | $75.36 \pm 0.44$ | $65.11 \pm 0.72$ | $79.70 \pm 0.59$ |
| SP+LEO (LOCAL) | $\mathbf{61.46 \pm 0.69}$ | $\mathbf{76.54 \pm 0.59}$ | $\mathbf{66.07 \pm 0.66}$ | $\mathbf{80.68 \pm 0.41}$ |
| LS META-LEARN | $60.19 \pm 0.65$ | $76.76 \pm 0.43$ | $64.32 \pm 0.65$ | $81.43 \pm 0.55$ |
| TASML (SP + LS META-LEARN) | $\mathbf{62.04 \pm 0.52}$ | $\mathbf{78.22 \pm 0.47}$ | $\mathbf{66.42 \pm 0.37}$ | $\mathbf{82.62 \pm 0.31}$ |

Tab. 2 compares TASML to a diverse set of previous methods, including MAML, R2D2, PRO-TONETS [47] and METAOPTNETS [29]. The results clearly show that TASML outperforms previous methods in both settings on the CIFAR-FS dataset, further validating the efficacy of the proposed structured prediction approach.

## 5.3 Improvements from Structured Prediction

The task-specific objective in (9) is model-agnostic, which enables us to leverage existing meta-learning methods, including architecture and optimization routines, to implement Alg. 1. For instance, we may replace LS META-LEARN in Sec. 4.1 with MAML, leading to a new instance of structured prediction-based conditional meta-learning.

Tab. 3 compares the average test accuracy of MAML, LEO and LS META-LEARN with their conditional counterparts (denoted SP + METHOD) under the proposed structured prediction perspective. For consistency with our earlier discussion, we use TASML to denote SP + LS META-LEARN, although the framework is generally applicable to most methods. We observe that the structured prediction variants consistently outperform the original algorithms in all experiment settings. In particular, our approach improves MAML by the largest amount, averaging ∼6% increase in test accuracy (e.g. from $48.70\%$ to $52.81\%$ for 5-way-1-shot on *mini*IMAGENET). TASML averages ∼ 1.5% improvements over LS META-LEARN. We highlight that the structured prediction improves also LEO – which is already a conditional meta-learning method – by ∼1%. This suggests that our structured prediction perspective is parallel to model-based conditional meta-learning, and might be combined to improve performance.

## 5.4 Model Efficiency

A potential limitation of TASML is the additional computational cost imposed by repeatedly performing the task-specific adaptation (9). Here we assess the trade-off between computations and accuracy improvement induced by this process. Fig. 2 reports the classification accuracy of TASML when minimizing the structured prediction functional in (9), with respect to the number of training steps $J$, starting from the initialization points ($J = 0$) learned via unconditional meta-learning using LS META-LEARN (refer to Tab. 3). Therefore Fig. 2 explicitly captures the additional performance improvements resulting from structured prediction. Aside from the slight decrease in performance on 1-shot *mini*IMAGENET after 50 steps, TASML shows a consistent and stable performance improvements over the 500 steps via structured prediction. The results present a trade-off between performance improvements over unconditional meta-learning, and additional computational costs from optimizing structured prediction functional in (9).

More concretely, we quantify the actual time spent on structured prediction steps for LS META-LEARN in Tab. 4, which reports the average number of meta-gradient steps per second on a single Nvidia GTX 2080. We note that 100 steps of structured prediction for LS META-LEARN – after which we observe the largest improvement in general – take about 6 seconds to complete. In addition, TASML takes ∼ $0.23s$ for computing $\alpha(D)$ given a meta-train set of 30k tasks. In applications where model accuracy has the priority, the additional computational cost is a reasonable trade-off for accuracy. The adaptation cost is also amortized overall future queries in the adapted model. Lastly,

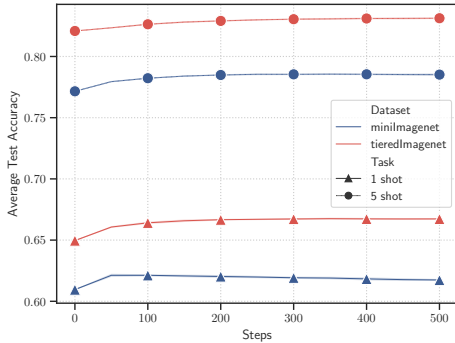

| (STEPS/SEC) | $mini$IMAGENET | $tiered$IMAGENET |
|---|---|---|
| LEO | $7.52 \pm 0.19$ | $6.95 \pm 0.47$ |
| TASML | $\mathbf{17.82 \pm 0.27}$ | $\mathbf{14.71 \pm 0.34}$ |

Table 4: Meta-gradient steps per second on 5-shot learning tasks

Figure 2: Average test performance over 500 TASML structured prediction steps.

we note that other conditional meta-learning methods also induce additional computational cost over unconditional formulation. For instance, Tab. 4 shows that LEO performs fewer meta-gradient steps per second during training, as it incurs more computational cost for learning conditional initialization.

### 5.5   Comparing Structured Prediction with Multi-Task Learning

We note that TASML's formulation (7) is related to objective functions for multi-task learning (MTL) that also learn to weight tasks (e.g. [12, 25]). However, these strategies have entirely different design goals: MTL aims to improve performance on *all* tasks, while meta-learning focuses only on the target task. This makes MTL unsuitable for meta-learning. To better visualize this fact, here we investigated whether MTL may be used as an alternative method to obtain task weighting for conditional meta-learning. We tested [25] on *mini*IMAGENET and observed that the method significantly underperforms TASML, achieving $56.8 \pm 1.4$ for 1-shot and $68.7 \pm 1.2$ for 5-shot setting. We also observe that the performance on target tasks fluctuated widely during training with the MTL objective function, since MTL does not prioritize the performance of the target task, nor prevent negative transfer towards it.

## 6   Conclusion and Future Works

We proposed a novel perspective on conditional meta-learning based on structured prediction. Within this context, we presented task-adaptive structured meta-learning (TASML), a general framework that connects intimately to existing meta-learning methods via task-specific objective functions. The proposed method is theoretically principled, and empirical evaluations demonstrated its efficacy and effectiveness compared to the state of the art. In future work, we aim to design parametric approaches for TASML to improve model efficiency. We also aim to investigate novel metrics that better capture the similarity between datasets, given the key role played by the kernel function in our framework.

## Broader Impact

Meta-learning aims to construct learning models capable of learning from experiences, Its intended users are thus primarily non-experts who require automated machine learning services, which may occur in a wide range of potential applications such as recommender systems and autoML. The authors do not expect the work to address or introduce any societal or ethical issues.

## Acknowledgments and Disclosure of Funding

The authors would like to thank the anonymous reviewers for their comments. This work was supported in part by UK DSTL/EPSRC Grant EP/P008461/1, National Science Scholarship from A*STAR Singapore, the Royal Academy of Engineering Chair in Emerging Technologies to Y.D. and the Royal Society of Engineering for grant SPREM RGS/R1/201149.

## Footnotes

[1]TASML implementation is available at https://github.com/RuohanW/Tasml

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
