[Supplementary Material]

# Supplementary Material: Structured Prediction for Conditional Meta-Learning

The Appendix is organized in two main parts:

- Appendix A proves the formal version of Theorem 1 and provides additional details on the connection between structured prediction and conditional meta-learning investigated in this work.

- Appendix B provides additional details on the model hyperparameters and additional experimental evaluation.

## A   Structured Prediction for Conditional Meta-learning

We first recall the general formulation of the structured prediction approach in [13], followed by showing how the conditional meta-learning problem introduced in Section 3 can be cast within this setting.

### A.1   General Structured Prediction

In this section, we borrow from the notation of [13]. Consider $\mathcal{X}, \mathcal{Y}$ and $\mathcal{Z}$ three spaces, respectively the *input*, *label* and *output* sets of our problem. We make a distinction between label and output space since conditional meta-learning can be formulated within the setting described below by taking $\mathcal{Z}$ to be the meta-parameter space $\Theta$ and $\mathcal{Y}$ the space $\mathcal{D}$ of datasets.

Structured prediction methods address supervised learning problems where the goal is to learn a function $f : \mathcal{X} \to \mathcal{Z}$ taking values in a "structured" space $\mathcal{Z}$. Here, the term structured is general and essentially encompasses output sets of strings, graphs, points on a manifold, probability distributions, etc. Formally, these are all spaces that are not linear or do not have a canonical embedding into a linear space $\mathbb{R}^k$.

As we will discuss in the following, the lack of linearity on $\mathcal{Z}$ poses concrete challenges on modeling and optimization. In contrast, formally, the target learning problem is cast as a standard supervised learning problem of the form (1). More precisely, given a distribution $\rho$ on $\mathcal{X} \times \mathcal{Y}$

$$\min_{f:\mathcal{X}\to\mathcal{Z}} \mathcal{E}(f) \qquad \text{with} \qquad \mathcal{E}(f) = \int \triangle(f(x), y|x) \, d\rho(x,y), \qquad \text{(A.1)}$$

where $\triangle : \mathcal{Z} \times \mathcal{Y} \times \mathcal{X} \to \mathbb{R}$ is a loss function measuring prediction errors. Note that $\triangle(z, y|x)$ does not only compare the predicted output $z \in \mathcal{Z}$ with the label $y \in \mathcal{Y}$, but does that also depending or *conditioned* on the input $x \in \mathcal{X}$ (hence the notation $\triangle(z, y|x)$ rather than $\triangle(z, y, x)$). These conditioned loss functions were originally introduced to account for structured prediction settings where prediction errors depend also on properties of the input. For instance in ranking problems or in sequence-to-sequence translation settings, as observed in [13].

**Structured Prediction Algorithm**[2]. Given a finite number $n \in \mathbb{N}$ of points $(x_i, y_i)_{i=1}^n$ independently sampled from $\rho$, the structured prediction algorithm proposed in [13] is an estimator $\hat{f} : \mathcal{X} \to \mathcal{Z}$ such that, for every $x \in \mathcal{X}$

$$\hat{f}(x) = \operatorname*{argmin}_{z\in\mathcal{Z}} \sum_{i=1}^n \alpha_i(x) \triangle(z, y_i|x_i). \qquad \text{(A.2)}$$

where, given a reproducing kernel $k : \mathcal{X} \times \mathcal{X} \to \mathbb{R}$, the weighs $\alpha$ are obtained as

$$\alpha(x) = (\alpha_1(x), \dots, \alpha_n(x))^\top \in \mathbb{R}^n \qquad \text{with} \qquad \alpha(x) = (\mathbf{K} + \lambda I)^{-1} v(x), \qquad \text{(A.3)}$$

where $\mathbf{K} \in \mathbb{R}^{n \times n}$ is the empirical kernel matrix with entries $K_{ij} = k(x_i, x_j)$ and $v(x) \in \mathbb{R}^n$ is the evaluation vector with entries $v(x)_i = k(x, x_i)$, for any $i, j = 1, \ldots, n$ and $\lambda > 0$ is a hyperparameter.

The estimator above has a similar form to the TASML algorithm proposed in this work in (7). In the following, we show that the latter is indeed a special case of (A.2).

## A.2 A Strucutred Prediction perspective on Conditional Meta-learning

In the conditional meta-learning setting introduced in Section 3 the goal is to learn a function $\tau : \mathcal{D} \to \Theta$ where $\mathcal{D}$ is a space of datasets and $\Theta$ a space of learning algorithms. We define the conditional meta-learning problem according to the expected risk (5) as

$$\min_{\tau : \mathcal{D} \to \Theta} \mathcal{E}(\tau) \qquad \text{with} \qquad \mathcal{E}(\tau) = \int \mathcal{L}\Big(\mathrm{Alg}\big(\tau(D^{tr}), D^{tr}\big), \ D^{val}\Big) \, d\pi(D^{tr}, D^{val}), \qquad \text{(A.4)}$$

where $\pi$ is a probability distribution sampling the pair of train and validation datasets $D^{tr}$ and $D^{val}$. We recall that the distribution $\pi$ samples the two datasets according to the process described in Section 2, namely by first sampling $\rho$ a task-distribution (on $\mathcal{X} \times \mathcal{Y}$) from $\mu$ and then obtaining $D^{tr}$ and $D^{val}$ by independently sampling points $(x, y)$ from $\rho$. Therfore $\pi = \pi_\mu$ can be seen as implicitly induced by $\mu$. In practice, we have only access to a meta-training set $S = (D_i^{tr}, D_i^{val})_{i=1}^N$ of train-validation pairs sampled from $\pi$.

We are ready to formulate the conditional meta-learning problem within the structured prediction setting introduced in Appendix A.1. In particular, we take the input and label spaces to correspond to the set $\mathcal{D}$ and choose as output set the space $\Theta$ of meta-parameters. In this setting, the loss function is of the form $\triangle : \Theta \times \mathcal{D} \times \mathcal{D} \to \mathbb{R}$ and corresponds to

$$\triangle(\theta, D^{val} | D^{tr}) = \mathcal{L}\Big(\mathrm{Alg}\big(\theta, D^{tr}\big), \ D^{val}\Big). \qquad \text{(A.5)}$$

Therefore, we can interpret the loss $\triangle$ as the function measuring the performance of a meta-parameter $\theta$ when the corresponding algorithm $\mathrm{Alg}(\theta, \cdot)$ is trained on $D^{tr}$ and then tested on $D^{val}$. Under this notation, it follows that (A.4) is a special case of the structured prediction problem (A.1). Therefore, casting the general structured prediction estimator (A.2) within this setting yields the TASML estimator proposed in this work and introduced in (7), namely $\tau_N : \mathcal{D} \to \Theta$ such that, for any dataset $D \in \mathcal{D}$,

$$\tau_N(D) = \operatorname*{argmin}_{\theta \in \Theta} \sum_{i=1}^N \alpha_i(D) \, \mathcal{L}\Big(\mathrm{Alg}\big(\theta, D^{tr}\big), \ D^{val}\Big),$$

where $\alpha : \mathcal{D} \to \mathbb{R}^N$ is learned according to (A.3), namely

$$\alpha(x) = (\alpha_1(x), \ldots, \alpha_N(x))^\top \in \mathbb{R}^N \qquad \text{with} \qquad \alpha(x) = (\mathbf{K} + \lambda I)^{-1} \, v(D),$$

with $\mathbf{K}$ and $v(D)$ defined as in (7). Hence, we have recovered $\tau_N$ as it was introduced in this work.

## A.3 Theoretical Analysis

In this section we prove Theorem A.1. Our result can be seen as a corollary of [Thm.5 14] applied to the generalized structured prediction setting of Appendix A.1. The result hinges on two regularity assumptions on the loss $\triangle$ and on the meta-distribution $\pi$ that we introduce below.

**Assumption 1.** *The loss $\triangle$ is of the form* (A.5) *and admits derivatives of any order, namely $\triangle \in C^\infty(\mathcal{Z} \times \mathcal{Y} \times \mathcal{X})$.*

Recall that by (A.5) we have

$$\mathcal{L}(\theta, D^{val}, D^{tr}) = \frac{1}{|D^{val}|} \sum_{(x,y) \in D^{val}} \ell\Big( \big[\mathrm{Alg}(\theta, D^{tr})\big](x), \ y \Big). \qquad \text{(A.6)}$$

Therefore, sufficient conditions for Assumption 1 to hold are: $i)$ the inner loss function $\ell$ is smooth (e.g. least-squares, as in this work) and $ii)$ the inner algorithm $\mathrm{Alg}(\cdot, \cdot)$ is smooth both with respect to the

meta-parameters $\theta$ and the training dataset $D^{tr}$. For instance, in this work, Assumption 1 is verified if the meta-representation network $\psi_\theta$ is smooth with respect to the meta-parametrization $\theta$. Indeed, $\ell$ is chosen to be the least-squares loss and the closed form solution $W(\theta, D^{tr}) = X_\theta^\top (X_\theta X_\theta^\top + \lambda I)^{-1} Y$ in (B.2) is smooth for any $\lambda > 0$.

The second assumption below concerns the regularity properties of the meta-distribution $\pi$ and its interaction with the loss $\triangle$. The assumption leverages the notion of Sobolev spaces. We recall that for a set $\mathcal{K} \subset \mathbb{R}^d$ the Sobolev space $W^{s,2}(\mathcal{K})$ is the Hilbert space of functions from $\mathcal{K}$ to $\mathbb{R}$ that have square integrable weak derivatives up to the order $s$. We recall that if $\mathcal{K}$ satisfies the cone condition, namely there exists a finite cone $C$ such that each $x \in \mathcal{K}$ is the vertex of a cone $C_x$ contained in $\mathcal{K}$ and congruent to $C$ [2, Def. 4.6], then for any $s > d/2$ the space $W^{s,2}(\mathcal{K})$ is a RKHS. This follows from the Sobolev embedding theorem [2, Thm. 4.12] and the properties of RKHS [see e.g. 8, for a detailed proof].

Given two Hilbert spaces $\mathcal{H}$ and $\mathcal{F}$, we denote by $\mathcal{H} \otimes \mathcal{F}$ the tensor product of $\mathcal{H}$ and $\mathcal{F}$. In particular, given two basis $(h_i)_{i \in \mathbb{N}}$ and $(f_j)_{j \in \mathbb{N}}$ for $\mathcal{H}$ and $\mathcal{F}$ respectively, we have

$$\langle h_i \otimes f_j, h_{i'} \otimes f_{j'} \rangle_{\mathcal{H} \otimes \mathcal{F}} = \langle h_i, h_{i'} \rangle_{\mathcal{H}} \cdot \langle f_j, f_{j'} \rangle_{\mathcal{F}},$$

for every $i, i', j, j' \in \mathbb{N}$. We recall that $\mathcal{H} \otimes \mathcal{F}$ is a Hilbert space and it is isometric to the space $\mathrm{HS}(\mathcal{F}, \mathcal{H})$ of Hilbert-Schmidt (linear) operators from $\mathcal{F}$ to $\mathcal{H}$ equipped with the standard Hilbert-Schmidt $\langle \cdot, \cdot \rangle_{\mathrm{HS}}$ dot product. In the following, we denote by $\mathsf{T} : \mathcal{H} \otimes \mathcal{F} \to \mathrm{HS}(\mathcal{F}, \mathcal{H})$ the isometry between the two spaces.

We are ready to state our second assumption.

**Assumption 2.** *Assume $\Theta \subset \mathbb{R}^{d_1}$ and $\mathcal{D} \subset \mathbb{R}^{d_2}$ compact sets satisfying the cone condition and assume that there exists a reproducing kernel $k : \mathcal{D} \times \mathcal{D} \to \mathbb{R}$ with associated RKHS $\mathcal{F}$ and $s > (d_1 + 2d_2)/2$ such that the function $g^* : \mathcal{D} \to \mathcal{H}$ with $\mathcal{H} = W^{s,2}(\Theta \times \mathcal{D})$, characterized by*

$$g^*(D^{tr}) = \int \triangle(\cdot, D^{val} | \cdot) \, d\pi(D^{val} | D^{tr}) \qquad \forall D^{tr} \in \mathcal{D}, \tag{A.7}$$

*is such that $g^* \in \mathcal{H} \otimes \mathcal{F}$ and, for any $D \in \mathcal{D}$, we have that the application of the operator $\mathsf{T}(g^*) : \mathcal{F} \to \mathcal{H}$ to the function $k(D, \cdot) \in \mathcal{F}$ is such that $\mathsf{T}(g^*) \, k(D, \cdot) = g^*(D)$.*

The function $g^*$ in (A.7) can be interpreted as capturing the interaction between $\triangle$ and the meta-distribution $\pi$. In particular, Assumption 2 imposes two main requirements: *i)* for any $D \in \mathcal{D}$ the output of $g^*$ is a vector in a Sobolev space (i.e. a function) of smoothness $s > (d_1 + 2d_2)/2$, namely $g^*(D) \in W^{s,2}(\Theta \times \mathcal{D})$ and, *ii)* we require $g^*$ to correspond to a vector in $W^{s,2}(\Theta \times \mathcal{D}) \otimes \mathcal{F}$. Note that the first requirement is always satisfied if Assumption 1 holds. The second assumption is standard in statistical learning theory [see e.g. 11, 46, and references therein] and can be interpreted as requiring the conditional probability $\pi(\cdot | D^{tr})$ to not vary dramatically for small perturbations of $D^{tr}$.

We are ready to state and prove our main theorem, whose informal version is reported in Theorem 1 in the main text.

**Theorem A.1** (Learning Rates). *Under Assumptions 1 and 2, let $S = (D_i^{tr}, D_i^{val})_{i=1}^N$ be a meta-training set of points independently sampled from a meta-distribution $\pi$. Let $\tau_N$ be the estimator in (7) trained with $\lambda_2 = N^{-1/2}$ on $S$. Then, for any $\delta \in (0, 1]$ the following holds with probability larger or equal than $1 - \delta$,*

$$\mathcal{E}(\tau_N) - \inf_{\tau : \mathcal{D} \to \Theta} \mathcal{E}(\tau) \leq c \log(1/\delta) \, N^{-1/4}, \tag{A.8}$$

*where $c$ is a constant depending on $\kappa^2 = \sup_{D \in \mathcal{D}} k(D, D)$ and $\|g^*\|_{\mathcal{H} \otimes \mathcal{F}}$ but independent of $N$ and $\delta$.*

*Proof.* Let $\mathcal{H} = W^{s,2}(\Theta \times \mathcal{D})$ and $\mathcal{G} = W^{s,2}(\mathcal{D})$. Since $s > (d_1 + 2d_2)/2$, both $\mathcal{G}$ and $\mathcal{H}$ are reproducing kernel Hilbert spaces (RKHS) [see discussion above or 8]. Let $\psi : \Theta \times \mathcal{D} \to \mathcal{H}$ and $\varphi : \mathcal{D} \to \mathcal{G}$ be two feature maps associated to $\mathcal{H}$ and $\mathcal{G}$ respectively. Without loss of generality, we can assume the two maps to be normalized.

We are in the hypotheses[3] of [32, Thm. 6 Appendix D], which guarantees the existence of a Hilbert-Schmidt operator $V : \mathcal{G} \to \mathcal{H}$, such that $\triangle$ can be characterized as

$$\triangle(\theta, D^{val} \mid D^{tr}) = \langle \psi(\Theta, D^{tr}), V\varphi(D^{val}) \rangle_{\mathcal{H}} \tag{A.9}$$

for any $D^{tr}, D^{val} \in \mathcal{D}$ and $\theta \in \Theta$. Since the feature maps $\varphi$ and $\psi$ are normalized [8], this implies also $\|V\|_{\mathrm{HS}} = \|\triangle\|_{s,2} < +\infty$, namely that the Sobolev norm of $\triangle$ in $W^{s,2}(\Theta \times \mathcal{D} \times \mathcal{D})$ is equal to the Hilbert-Schmidt norm of $V$.

The result in (A.9) corresponds to the definition of *Structure Encoding Loss Function (SELF)* in [13, Def. 1]. Additionally, if we denote $\widetilde{\varphi} = V\varphi$, we obtain the equality

$$g^*(D^{tr}) = \int \widetilde{\varphi}(D^{val}) \, d\pi(D^{val}|D^{tr}) = \int \triangle(\cdot, D^{val}|\cdot) \, d\pi(D^{val}|D^{tr}), \tag{A.10}$$

for all $D^{tr} \in \mathcal{D}$, where $g^* : \mathcal{D} \to \mathcal{H}$ is defined as in Assumption 2, we are in the hypotheses of the comparison inequality theorem [13, Thm. 9]. In our setting, this result states that for any measurable function $g : \mathcal{D} \to \mathcal{H}$ and the corresponding function $\tau_g : \mathcal{D} \to \Theta$ defined as

$$\tau_g(D) = \operatorname*{argmin}_{\theta \in \Theta} \ \langle \psi(\theta, D), g(D) \rangle_{\mathcal{H}} \qquad \forall D \in \mathcal{D}, \tag{A.11}$$

we have

$$\mathcal{E}(\tau_g) - \inf_{\tau:\mathcal{D}\to\Theta} \mathcal{E}(\tau) \ \leq \ \sqrt{\int \|g(D) - g^*(D)\|_{\mathcal{H}}^2 \ d\pi_{\mathcal{D}}(D)}, \tag{A.12}$$

where $\pi_{\mathcal{D}}(D^{tr})$ denotes the marginal of $\pi(D^{val}, D^{tr})$ with respect to training data. Note that the constant $\mathsf{c}_{\triangle}$ that appears in the original comparison inequality is upper bounded by 1 in our setting since $\mathsf{c}_{\triangle} = \sup_{D,\theta} \|\psi(\theta, D)\|$ and the feature map $\psi$ is normalized.

Let now $g_N : \mathcal{D} \to \Theta$ be the minimizer of the vector-valued least-squares empirical risk minimization problem

$$g_N = \operatorname*{argmin}_{g \in \mathcal{H} \otimes \mathcal{F}} \ \frac{1}{N} \sum_{i=1}^{N} \ \left\| g(D_i^{tr}) - \tilde{\varphi}(D_i^{val}) \right\|_{\mathcal{H}}^2 + \lambda_2 \|g\|_{\mathcal{H} \otimes \mathcal{F}}^2 \ .$$

This problem can be solved in closed form and it can be shown [14, Lemma B.4] that $g_N$ is of the form

$$g_N(D) = \sum_{i=1}^{n} \alpha_i(D) \, \tilde{\varphi}(D_i^{val}), \tag{A.13}$$

for all $D \in \mathcal{D}$, where $\alpha_i(D)$ is defined as in (7). Due to linearity [see also Lemma 8 in 13] we have

$$\tau_{g_N}(D) = \operatorname*{argmin}_{\theta \in \Theta} \ \langle \psi(\theta, D), g_N(D) \rangle_{\mathcal{H}} \tag{A.14}$$

$$= \operatorname*{argmin}_{\theta \in \Theta} \ \sum_{i=1}^{N} \alpha_i(D) \, \mathcal{L}\Big( \mathrm{Alg}(\theta, D^{tr}), \ D^{val} \Big) \tag{A.15}$$

$$= \tau_N(D), \tag{A.16}$$

which corresponds to the estimator $\tau_N(D)$ studied in this work and introduced in (7). The comparison inequality (A.12) above, becomes

$$\mathcal{E}(\tau_N) - \inf_{\tau:\mathcal{D}\to\Theta} \mathcal{E}(f) \ \leq \ \sqrt{\int \|g_N(D) - g^*(D)\|_{\mathcal{H}}^2 \ d\pi_{\mathcal{D}}(D)}. \tag{A.17}$$

Therefore, we can obtain a learning rate for the excess risk of $\tau_N$ by studying how well the vector-valued least-squares estimator $g_N$ is approximating $g^*$. Since $g^* \in \mathcal{H} \otimes \mathcal{F}$ from the hypothesis, we can replicate the proof in [14, Thm. 5] to obtain the desired result. Note that by framing our problem in such context we obtain a constant $c$ that depends only on the norm of $g^*$ as a vector in $\mathcal{H} \otimes \mathcal{F}$. We recall that $g^*$ captures the "regularity" of the meta-learning problem. Therefore, the more regular (i.e. easier) the learning problem, the faster the learning rate of the proposed estimator. $\qquad\square$

# B   Model and Experiment Details

We provide additional details on the model architecture, experiment setups, and hyperparameter choices. We performed only limited mode tuning, as it is not the focus on the work.

## B.1   Details on LS META-LEARN

Meta-representation learning methods formulate meta-learning as the process of finding a shared representation to be fine-tuned for each task. Formally, they model the task predictor as a composite function $f_W \circ \psi_\theta : \mathcal{X} \to \mathcal{Y}$, with $\psi_\theta : \mathcal{X} \to \mathbb{R}^p$ a shared feature extractor meta-parametrized by $\theta$, and $f_W : \mathbb{R}^p \to \mathcal{Y}$ a map parametrized by $W$. The parameters $W$ are learned for each task as a function $W(\theta, D)$ via the inner algorithm

$$f_{W(\theta,D)} \circ \psi_\theta \ = \ \mathrm{Alg}(\theta, D). \tag{B.1}$$

[9] proposed $\mathrm{Alg}(\theta, D)$ to perform empirical risk minimization of $f_W$ over $D = (x_i, y_i)_{i=1}^m$ with respect to the least-squares loss $\ell(y, y') = \|y - y'\|^2$. Assuming[4] $\mathcal{Y} = \mathbb{R}^C$ and a linear model for $f_W$, this corresponds to performing ridge-regression on the features $\psi_\theta$, yielding the closed-form solution

$$W(\theta, D) \ = \ X_\theta^\top (X_\theta X_\theta^\top + \lambda_\theta I)^{-1} Y, \tag{B.2}$$

where $\lambda_1 > 0$ is a regularizer. $X_\theta \in \mathbb{R}^{m \times p}$ and $Y \in \mathbb{R}^{m \times C}$ are matrices with $i$-th row corresponding to the $i$-th training input $\psi_\theta(x_i)$ and output $y_i$ in the dataset $D$, respectively. The closed-form solution (B.2) has the advantage of being $i$) efficient to compute and $ii$) suited for the computation of meta-gradients with respect to $\theta$. Indeed, $\nabla_\theta W(\theta, D)$ can be computed in closed-form or via automatic differentiation.

LS META-LEARN is a meta-representation learning algorithm consists of:

- The *meta-representation* architecture $\psi_\theta : \mathcal{X} \to \mathbb{R}^p$ is a two-layer fully-connected network with residual connection [21].
- The *task predictor* $f_W : \mathbb{R}^p \to \mathcal{Y}$ is a linear model $f_W\big(\psi_\theta(x)\big) = W\psi_\theta(x)$ with $W \in \mathbb{R}^{C \times p}$ the model parameters. We assume $\mathcal{Y} = \mathbb{R}^C$ (e.g. one-hot encoding of $C$ classes in classification settings).
- The *inner algorithm* is $f_{W(\theta,D)} \circ \psi_\theta = \mathrm{Alg}(\theta, D)$, where $W(\theta, D)$ is the least-squares closed-form solution introduced in (B.2).

We note that [9] uses the cross-entropy $\ell$ to induce $\mathcal{L}$. Consequently, when optimizing the meta-parameters $\theta$, the performance of $W(\theta, D)$ is measured on a validation set $D'$ with respect to a loss function (cross-entropy) different from the one used to learn it (least-squares). We observe that such incoherence between inner- and meta-problems lead to worse performance than least-square task loss.

## B.2   Model Architecture

Given the pre-trained representation $\varphi(x) \in \mathbb{R}^{640}$, the proposed model is $f_\theta(\varphi(x)) = \varphi(x) + g_\theta(\varphi(x))$, a residual network with fully-connected layers. Each layer of the fully-connected network $g_\theta(\varphi(x))$ is also 640 in dimension.

We added a $\ell_2$ regularization term on $\theta$, with a weight of $\lambda_\theta$ reported below.

For top-$M$ values from $\alpha(D)$, we normalize the values such that they sum to 1.

## B.3   Experiment Setups

We use the same experiment setup as LEO [44] by adapting its official implementation[5]. For both 5-way-1-shot and 5-way-5-shot settings, we use the default environment values from the implementation, including a meta-batch size of 12, and 15 examples per class for each class in $D^{val}$ to ensure a fair comparison.

## B.4  Model Hyperparameters

Models across all settings share the same hyperparameters, listed in Table 5.

Table 5: Hyperparameter values used in the experiments

| SYMBOL | DESCRIPTION | VALUES |
|---|---|---|
| $\lambda$ IN (7) | REGULARIZER FOR LEARNING $\alpha(D)$ | $10^{-8}$ |
| $\lambda_\theta$ IN (B.2) | REGULARIZER FOR THE LEAST-SQUARE SOLVER, | 0.1 |
| $\sigma$ IN (10) | KERNEL BANDWIDTH | 50 |
| $\eta$ | META LEARNING RATE | $10^{-4}$ |
| $N$ | TOTAL NUMBER OF META-TRAINING TASKS | $30,000$ |
| $M$ | NUMBER OF TASKS TO KEEP IN ALGORITHM 1 | 500 |

## C  Additional Ablation Study

### C.1  Structured Prediction from Random Initialization

We note that the unconditional initialization of $\theta$ from Section 4 is optional and designed for improving computational efficiency. Figure 3 reports how TASML performs, starting from random initialization. The results suggest that structured prediction takes longer to converge with random initialization, but achieves performance comparable to Table 1. In addition, structured prediction appears to work well, despite having access to only a small percentage of meta-training tasks ($M = 1000$ in this experiment).

Figure 3: Average test task performance over 1000 structured prediction steps. Structured prediction takes longer to converge from random initialization, but achieves comparable performance.

### C.2  Top-M Filtering

Table 6 reports the performance of structured prediction by varying the number $M$ of tasks used. We use 5-way-5-shot on *mini*IMAGENET and *tiered*IMAGENET as the experiment settings.

The results show that TASML is robust to the choice of $M$. As $M$ increases, its impact on performance is small as most tasks have tiny weights with respect to the objective function.

| $M$ | 100 | 500 | 1000 | 10000 | 30000 |
|---|---|---|---|---|---|
| *mini*IMAGENET (%) | $77.60 \pm 0.30$ | $78.22 \pm 0.47$ | $78.43 \pm 0.39$ | $78.47 \pm 0.37$ | $78.51 \pm 0.42$ |
| *tiered*IMAGENET (%) | $81.95 \pm 0.23$ | $82.62 \pm 0.31$ | $82.95 \pm 0.27$ | $83.01 \pm 0.29$ | $83.03 \pm 0.35$ |

Table 6: The effects of top-$M$ filtering on structured prediction accuracy. TASML is robust to the choices of $M$.

## C.3 Explicit Dependence on Target Tasks

In (9), we introduced an additional $\mathcal{L}\big(\text{Alg}(\theta, D), D\big)$, such that the objective function explicitly depends on target task $D$. To study the contribution of each term towards test accuracy, we modify (9) by weighting the contribution of each term by the weights $\beta_1, \beta_2 \geq 0$

$$\tau(D) = \underset{\theta \in \Theta}{\operatorname{argmin}} \ \beta_1 \sum_{i=1}^{N} \alpha_i(D) \, \mathcal{L}\big(\text{Alg}(\theta, D_i^{tr}), \ D_i^{val}\big) + \beta_2 \mathcal{L}\big(\text{Alg}(\theta, D), \ D\big), \qquad \text{(C.1)}$$

We report the test accuracy on *mini*IMAGENET and *tiered*IMAGENET on 5-way-5-shot below.

| | $\beta_1 = 0, \beta_2 = 1$ | $\beta_1 = 1, \beta_2 = 0$ | $\beta_1 = 1, \beta_2 = 1$ | $\beta_1 = 1, \beta_2 = 2$ |
|---|---|---|---|---|
| *mini*IMAGENET (%) | $73.59 \pm 0.49$ | $77.32 \pm 0.36$ | $78.22 \pm 0.47$ | $78.51 \pm 0.32$ |
| *tiered*IMAGENET (%) | $79.74 \pm 0.62$ | $81.63 \pm 0.47$ | $82.62 \pm 0.31$ | $83.01 \pm 0.43$ |

Table 7: Test accuracy on *mini*IMAGENET and *tiered*IMAGENET by adjusting the importance of each term in (C.1)

Table 7 suggests that the explicit dependence on target task $D$, **combined with** other relevant tasks, provides the best training signal for TASML. In particular, optimizing with respect to the target task alone (i.e. $\beta_1 = 0, \beta_2 = 1$) leads to overfitting while excluding the target task (i.e. $\beta_1 = 1, \beta_2 = 0$) ignores valuable training signal, leading to underfitting. Ultimately, both extremes lead to sub-optimal performance. The results in Table 7 show that both terms in (9) are necessary to achieve good test accuracy.

## C.4 Choice of Kernel for Structured Prediction

Lastly, we study how the choice of kernel from (10) affects test accuracy. In addition to the Gaussian kernel considered in this work, we include both the linear kernel

$$k(D, D') = \langle \Phi(D), \Phi(D') \rangle + c$$

with $c > 0$ a hyperparameter, and the Laplace kernel

$$k(D, D') = \exp(-\left\| \Phi(D) - \Phi(D') \right\| / \sigma)$$

(a) *mini*IMAGENET          (b) *tiered*IMAGENET

Figure 4: Test performance with different choices of kernel for structured prediction. Gaussian kernel obtains the best performance

with $\sigma > 0$ a hyperparameter. We considered the 5-way-5-shot task on *mini*IMAGENET and *tiered*IMAGENET to compare the impact of the kernel on TASML.

Figure 4 shows that the Gaussian kernel overall obtains the best performance among the three candidates. In our experiments, we observed that Gaussian kernels are most robust with respect to bandwidth parameters, while Laplace kernels appeared sensitive to the bandwidth parameter. Careful model selection for the bandwidth parameter might lead to better or comparable performance, but it is beyond the scope of this work. In addition, we observed the linear kernel to perform well in some settings but less expressive in general.

## Footnotes

[2]We note that in the original work, the authors considered a further parametrization of the loss $\triangle$ leveraging the concept of locality and parts. This led to the derivation of a more general (and involved) characterization of the estimator $\hat{f}$. However, for the setting considered in this work we consider a simplified scenario (see Appendix A.2 below) and we can therefore restrict to the case where the loss does not assume a factorization into parts, namely the set of parts $P$ corresponds to $P = \{1\}$ the singleton, leading to the structured prediction estimator (A.2).

[3]the original theorem was applied to the case where $\mathcal{Z} \times \mathcal{X} = \mathcal{Y}$ was the probability simplex in finite dimension. However the proof of such result requires only that $\mathcal{H}$ and $\mathcal{G}$ are RKHS and can therefore be applied to the general case where $\mathcal{Z} \times \mathcal{X}$ and $\mathcal{Y}$ are different from each other and they do not correspond to the probability simplex but are rather subset of $\mathbb{R}^k$ (possibly with different dimension for each space) and satisfy the boundary condition [8]. Therefore in our setting we can take $\mathcal{Z} = \Theta$ and $\mathcal{X} = \mathcal{Y} = \mathcal{D}$ to obtain the desired result.

[4]For instance, $C$ is the total number of classes, and $y \in \mathcal{Y}$ the one-hot encoding of a class in classification tasks

[5]https://github.com/deepmind/leo