[Reviews · NeurIPS 2020]

Review 1

Summary and Contributions: The authors present a new meta-learning framework in this paper, which involves meta-training tasks during the meta-testing process. In this framework, for each meta-testing task, it selects the most relevant K meta-training tasks by calculating kernel-based weight and construct a weighted loss. By optimizing the weighted loss, the framework achieves better performance on both miniImagenet and tieredImagenet.

Strengths: (1) The discussion about task conditioning in meta-learning is interesting and important. (2) The experiments are comprehensive.

Weaknesses: Thanks the authors for addressing some of my previous comments, with more clear descriptions and experimental comparison. Especially, more task conditioning methods (e.g., MMAML) are considered in this paper. However, my major concern has not been addressed. (1) The main criticism is about the setting and discussion with multi-task learning, which is also a major concern in my previous comments. The authors still ignore the discussion with multi-task learning. From my perspective, the goal for meta-learning is to generalize knowledge from previous tasks, which further benefits the training of a new task. The setting in this paper allows a new meta-testing task to access all meta-training tasks. In this way, a task memory (covers all tasks) is constructed in the meta-learning process. The improvement of structured prediction (Table 2) is mainly achieved by explicitly involving more tasks during meta-testing, which is totally different from other meta-learning settings, including gradient-based meta-learning and non-parametric meta-learning. If the authors involve all meta-training tasks, the setting is more like: we have one major task (i.e., the meta-testing task) and a lot of auxiliary tasks (i.e., meta-training tasks). It is very similar to multi-task learning, it would be more convincing if the authors can discuss and compare some multi-task learning algorithms (e.g., [1,2]). Based on the above discussion, in Remark 1, the authors build the connection between MAML and TASML. It is correct for Eq. (7) since MAML is an unweighted algorithm. However, in the empirical practice (i.e., Eq. (9)), MAML is a totally different algorithm. Could the authors provide more discussion about the Eq. (9) and MAML? (2) For the efficiency comparison (i.e., Table 3), is the kernel computation time included? Since almost all gradient-based meta-learning algorithms do not include meta-training tasks during meta-testing time, for fair comparison, could the authors please provide the cost for kernel calculation (i.e., line Compute task weight \alpha(D) in Algorithm 1)? (3) The model architecture may be different from the model structure in MAML and LEO. The 48.70% performance (5-way, 1-shot miniImagenet) of MAML is achieved by the 4 convolutional layers + 1 fully connected classifier. For LEO, I checked the original paper, they use WRN-28-10 as backbone. Since the description in B.2 overlooks some details. Is the model structure used in this paper the same as LEO? Minor: one recent task conditioning paper in meta-learning could be included in future version [3]. [1] Chen, Zhao, et al. "GradNorm: Gradient Normalization for Adaptive Loss Balancing in Deep Multitask Networks." International Conference on Machine Learning. 2018. [2] Kendall, Alex, Yarin Gal, and Roberto Cipolla. "Multi-task learning using uncertainty to weigh losses for scene geometry and semantics." Proceedings of the IEEE conference on computer vision and pattern recognition. 2018. [3] Lee, Hae Beom, Hayeon Lee, Donghyun Na, Saehoon Kim, Minseop Park, Eunho Yang, and Sung Ju Hwang. "Learning to Balance: Bayesian Meta-Learning for Imbalanced and Out-of-distribution Tasks." ICLR 2020. # After rebuttal The authors response has addressed some of my concerns. I hope the authors could revise the paper according to the review. Thus, I increase my score.

Correctness: Remarker 1 seems inconsistent with the practical implementation.

Clarity: Yes, the paper is well written.

Relation to Prior Work: The paper ignores the discussion with multi-task learning.

Reproducibility: Yes

Additional Feedback:


Review 2

Summary and Contributions: The paper proposes a new algorithm for conditional meta-learning by casting the problem as a particular instance of structure prediction. More specifically, the authors focus on a particular class of meta-learning algorithms, i.e. meta-learning as a nested optimisation problem, and consider the structure output to be the inner learning algorithm parameterised by the meta-parameters. In particular, they resort to a recently proposed algorithm for structure prediction that, at meta-test time, weights the meta-loss for each task according to how similar the new task is to the tasks observed at training time. Finally, the authors provide theoretical guarantees and benchmark the method showing that outperforms state-of-the-art competitors. I am incline to accept the paper (the idea is simple and well motivated, the literature review thorough, results are competitive and accompanied by theoretical guarantees).

Strengths: - It is a natural extension of structure prediction algorithms yet novel, establish tight connections between two different areas (structure prediction and meta-learning). I think the simplicity of the idea is one of the biggest strengths of the paper. - The proposed algorithm can be easily applied to any unconditional meta-learning based on nested optimisation. This is a really attractive property from the practical point of view. - I comes with theoretical guarantees. - The paper is very well written and motivated

Weaknesses: - The theoretical guarantees are derived without including the practical elements of the algorithm, e.g. the last term in eq. (9) is left out. - It is nonparametric, and so it needs access to the training set of all tasks at test time. It is unclear to me how to apply it in mini-batch settings since it needs to compute the kernel across all datasets in advance - The experimental results are good, although could be strengthen by adding some additional benchmarks, e.g. CIFAR-FS. In addition, the last term in eq. (9) seems to be quite important in beating the performance of previous meta-learning algorithms (looking at the ablation study in the appendix). I am missing a more well-thought motivation in the main manuscript of this term.

Correctness: I have reviewed the equations of the paper and supplementary material and I have not caught any error. However, I am not an expert so I may have missed something, especially in the proof of the theorem.

Clarity: Yes

Relation to Prior Work: The authors make a good work at presenting the existing literature in both fields (meta-learning and structure prediction) and analysing the connections between their methods and existing ones. In particular, I really enjoyed reading Appendix A.1 and I think the paper would benefit from bringing back some parts into the main manuscript.

Reproducibility: Yes

Additional Feedback: Clarity: - Figure 1 could be improved. It was difficult to understand what the authors tried to convey when I first looked at it. - The connection with fairness in the broader impact statements comes out of the blue and there is not motivation. Typos: - Manuscript (line 28): to find a common set of meta-parameters - Appendix: I will replace the refs to the equations in the manuscript with the actual equations to improve readability. - Appendix (line 499): Sobolev embedding theorem - Appendix (line 622): with respect to the - Appendix (line 533): The reference seems to be wrong. Questions: - (Appendix 578): Are the parameters of the meta-representation learnt as well? - (Manuscript Table 3): I really appreciate the authors including this figure to address concerns about efficiency of their non-parametric model. However, I would appreciate a more detailed analysis in terms of computational complexity to better understand why these two numbers are so different and how the different approximations factor in.


Review 3

Summary and Contributions: This work views conditional meta learning as a structured prediction problem. They derive a non parametric framework for conditional meta-learning by weighting meta-training data appropriately for test time tasks. This can be combined with existing meta-learners to improve few shot classification performance. They provide some theoretical guarantees about algorithm performance and generalization. The authors motivate conditional meta-learning as being consistently better than unconditional variants, but lacking theoretical guarantees and requiring hand defined architectures. The key insight that they make is that they can interpret the inner loop of meta learning as a structured output to be predicted given some target tasks. This allows this to derive a principled estimator which ends up weighting training tasks non parametrically wrt the desired test task. They then introduce a specific practical algorithm which can be run on realistic datasets. I like this line of intuition they provide “Intuitively, the proposed framework learns a target task by explicitly recalling only the most relevant tasks from past observations, to better capture the local tasks distribution for improved generalization”

Strengths: I think the paper has statistically significant improvement over prior works in terms of meta-testing accuracy, not that much worse computational efficiency. The algorithm introduces an interesting idea of choosing the meta-training tasks that are most relevant to the test time task, which is more generally applicable. The algorithm has theoretical guarantees and the practical approximations are generally well motivated.

Weaknesses: There are some portions which are a bit hard to understand clarity wise (mostly Section 3), and a few gaps in motivation for the method. I think having a few more domains for comparison experimentally would also be helpful. I think the algorithm does induce extra computational cost since it is non parametric, as opposed to standard methods but I'm not sure that is a deal breaker.

Correctness: As far as a I can tell the proofs and claims are correct. The empirical methodology seems standard.

Clarity: It is mostly written alright, but I think there are some gaps in motivation as discussed in my detailed comments. Some portions have motivation jumps - eg why structured prediction solves problems with conditional meta-learning, etc. I also think Section 3 could have some more clarity into what is happening at training time, what is happening at test time, and why this is better than alternatives.

Relation to Prior Work: It is quite clearly discussed how this method compares to unconditional meta-learning, conditional meta-learning algorithms like [Jherfel et al], structured prediction approaches, etc. I think this is quite distinct from the prior works and well done.

Reproducibility: Yes

Additional Feedback: # After Rebuttal I think the author feedback was quite thorough and addressed my questions, so I will keep my score at an accept. This work views conditional meta learning as a structured prediction problem. They derive a non parametric framework for conditional meta-learning by weighting meta-training data appropriately for test time tasks. This can be combined with existing meta-learners to improve few shot classification performance. They provide some theoretical guarantees about algorithm performance and generalization. The authors motivate conditional meta-learning as being consistently better than unconditional variants, but lacking theoretical guarantees and requiring hand defined architectures. The key insight that they make is that they can interpret the inner loop of meta learning as a structured output to be predicted given some target tasks. This allows this to derive a principled estimator which ends up weighting training tasks non parametrically wrt the desired test task. They then introduce a specific practical algorithm which can be run on realistic datasets. I like this line of intuition they provide “Intuitively, the proposed framework learns a target task by explicitly recalling only the most relevant tasks from past observations, to better capture the local tasks distribution for improved generalization” This premise totally makes sense as a means to do meta-training only on tasks that are most relevant to the desired test task. It’s not immediately clear to me why the insight of this being structured prediction is necessary to implement the intuition described above though. Maybe the authors could clarify this? They show that their algorithm improves test time few shot accuracy, while still being quite efficient on new test time problems. The background is clear and well written. This point “However, existing methods typically require handcrafted architectures,” is quite unclear. Can the authors please elaborate? I think the conditional meta-learning subsection in Section 3 can be moved to preliminaries. It is a bit confusing to put it in the methods section, since it is just describing formalism from prior work. This statement “Intuitively, we can expect a significant improvement from the solution τ∗ of (5) compared to the solution θ∗ of (2), since by construction E(τ∗) ≤ E(θ∗)” is not clear. Why is this? I find there is a significant motivation gap going from “conditional meta-learning is good but needs handcrafted architectures to saying that viewing meta-learning as structured prediction will actually fix this issue”. Why? The authors talk about “structure encoding loss function” as a well known method. Could they please provide some background on this for readers from meta-learning so the paper is self contained? Maybe in the background section. The algorithm in 7 is a bit confusing. This seems like it is actually what happens at meta-test time on a new task. This should be mentioned very clearly. It should also answer the question of what is done at meta-training time then. Or is all the learning done non-parametrically at meta-test time. It’s a bit confusing what is done offline, what is done for the new datapoint/dataset, etc. From algorithm 1, it seems like a lot of the learning is done at meta-test time on a new task so this can get quite expensive since we are no longer amortizing the cost of meta-learning. How do the authors see this as a problem while learning? Also how important is the choice of Kernel to learning? It seems like this problem is addressed by warmstarting from unconditional meta-learning, but does that affect the learning rates/asymptotic guarantees from earlier? Section 4 is very useful in explaining the design choices of the algorithm and motivates a useful practical algorithm. Are there any downsides to using the closed form least squares solver for task loss as opposed to a more expressive, standard loss function for meta-learning such as the one used in MAML. It would be nice if the authors could in the experiments explicitly state and outline which portions of their many proposed improvements really matter and which parts are just nice to have. Like what really makes it tick? The results seem pretty good compared to many standard meta-learning baselines. Could the results also be tried on other meta-learning datasets beyond imagenet? Maybe something like omniglot or CIFAR style datasets might also be worth considering. Section 5.3 does a good job of addressing my concerns about the model efficiency because of the requirement for test-time task specific adaptation.


Review 4

Summary and Contributions: This paper applies the idea called “structured prediction” in the domain of meta-learning. Relevant tasks from the meta-training stage are picked for the target task and can potentially alleviate the challenge of few-shot. Experiments show improvements compared to baselines.

Strengths: The theoretical ground is complete and the empirical results show improvement on 2 datasets compared to baselines and analyze the efficiency of the proposed model. The problem is relevant to NeurIPS community.

Weaknesses: I have a concern of the term: structured meta-learning, proposed in this paper. To my understanding, this term looks like the model is utilizing structural information, such as class hierarchy graph [1] and learned task hierarchy [2]. This may cause some confusion especially when this meta-learning community has already introduced related terms. Another concern is that the proposed method is training on the support set for every target task. Even if the efficiency has been discussed in this paper, there’s still concerns: far every task the model needs to “retrieve” tasks from a pool of tasks and this pool can be large (and can be unavailable) thus needs lot of memory, which is infeasible when applying in practice. [1] Learning to Propagate for Graph Meta-Learning [2] Hierarchically structured meta-learning

Correctness: I would suggest the authors to include the inference time comparison when comparing to baselines.

Clarity: In general, good. The clarity and format can be improved. And the use of term may raise confusion.

Relation to Prior Work: Although this work has a section of background and notation, they only include some classical methods in the field of meta-learning and does not discussed the most related works, such as [1,2,3,4,5]. [1,2] utilize structures for meta-learning. [3] use the augmented memory but stored different elements in the memory. [4,5] build task-aware features for each task. [3] Meta-learning with memory-augmented neural networks. [4] TADAM: Task dependent adaptive metric for improved few-shot learning [5] Task-Aware Feature Embeddings for Low Shot Learning

Reproducibility: Yes

Additional Feedback: Can I ask how do you split methods into conditional and unconditional in Table 1? Is there any definition?

[Author Response · NeurIPS 2020]

**General:** we thank the reviewers for their valuable feedback. We first address questions shared by most reviewers:

• **Alg. 1.** We clarify what happens during meta-train and -test respectively for Alg. 1. *Meta-train:* Alg. 1 inverses
the regularized kernel matrix $(\mathbf{K} + \lambda I)^{-1}$, costing $O(N^3)$ for $N$ meta-train tasks. Unconditional meta-learning is
recommended (not mandatory, see App. C.1) as a warm start for TASML. *Meta-test:* The kernel vector $v(D)$ and
weights $\alpha(D)$ are computed, costing $O(N^2)$ operations. $\alpha(D)$ are used in eq. (7) (or 9) to perform adaptation.

• **Inference Time.** For model adaptation, TASML takes $\sim 0.23s$ (computing $\alpha(D)$ for $N = 30k$) and $\sim 6s$ (see line
290) optimizing (9). In applications where model accuracy has the priority (e.g. AutoML services), it can be reasonable
to trade-off time for accuracy. The adaptation cost is also amortized over all future queries in the adapted model.

• **Experiments on CIFAR-FS.** We chose the same settings as those used to obtain Tab. 2 in our paper. For 1-shot,
TASML (**74.6 ± 0.7**), Leo ($71.2 \pm 0.6$), and MAML ($68.8 \pm 0.7$). For 5-shot, TASML (**85.1 ± 0.4**), Leo ($82.0 \pm 0.4$),
and MAML ($83.7 \pm 0.7$). TASML significantly outperforms the baselines, in line with findings in the paper.

**R2.** *Multi-task Learning (MTL)* While MTL may be used for meta-learning as a heuristic, it does not prioritize
performance of target tasks, nor prevent negative transfer towards it. In contrast, TASML only selects the most relevant
tasks for adaptation in a principled way. We implemented Kendall et. al on miniImagnet. The results are $56.8 \pm 1.4$
(1-shot) and $68.7 \pm 1.2$ (5-shot), under-performing TASML. Critically, each target task's performance swing widely
when trained with the MTL loss, which makes learning unstable. Our additional experiments suggest negative transfer
as a main issue with applying MTL to meta-learning.

• *Eq. (9) and MAML.* Both Eq. (7) and (9) are task-specific objectives with near-identical implementation. (9) is a
variant of (7), where (9) also exploits (few) labeled samples from target task during training. Remark 1 applies to both.
• *Different architectures in baselines.* Tab. 1 cited results from previous papers. For fairness, Tab. 2 reports results for
MAML with WRN-28-10 (i.e. LEO's feature), and that structured prediction can improve both MAML and LEO.

**R4. Clarify** (9)**.** The additional term can be interpreted as a special task where support and query sets coincide. The
term regularizes models to focus on relevant features from selected tasks, in order to perform well on target tasks.

• *Mini-batches and kernel evaluation.* For each target task $D$, we first compute $\alpha(D)$ against the entire meta-train set
(see **Alg. 1** and **Inference Time** above). Each mini-batch samples $k$ tasks and their weights from $M$-filtered meta-train
set to optimize eq. (9) restricted to the mini-batch.
• *Whether meta-representation is learned* Yes. the parameters of the meta-representation are learned.
• *Tab. 3.* LEO is slower during meta-train due to network complexity and having to learn task-conditional initialization.
TASML's network is simpler and more efficient to train, but diverts task-conditioning to test time (see **Inference Time**).

**R5.** *Is structured prediction (SP) necessary?* SP is not the only way to formalize the problem, and our paper reviewed
several existing conditional meta-learning methods. Rather, SP offers a principled strategy for conditional meta-learning,
for which we can study the statistical properties. These qualities make such perspective appealing.

• *"Handcrafted architecture" and motivation gap.* We will improve the phrasing: for most previous conditional
methods, the network design is ad-hoc to implement the specific conditional principles (e.g. task clustering). In contrast,
TASML uses kernel to implicitly captures task similarity, and yields weighted loss functions, which are more likely to
generalize to different application settings, and augments existing methods (see Tab. 2).
• *On the inequality $\mathcal{E}(\tau_*) \leq \mathcal{E}(\theta_*)$.* Conditional meta-learning minimizes $\mathcal{E}(\cdot)$ over $\mathcal{T}$ (all measurable functions
$\tau : \mathcal{D} \to \Theta$), a significantly larger set than $\Theta$ (all *constant* functions from $\mathcal{D}$ to $\Theta$). Hence $\min_{\mathcal{T}} \mathcal{E}(\tau) \leq \min_{\Theta} \mathcal{E}(\theta)$.
• *Does warm-start affect rates?* No, Thm.1 does not make assumptions about the initial model parameters.
• *Choice of Kernel.* We compared and discussed the impact of different kernels options in App. C.4.
• *Using least-squares (LS) loss* For few-shot classification task, there appears to be no drawback in our experiments.
On the contrary, LS enables efficient meta-gradient computation and speed up learning.
• *What makes TASML "tick"?* Top-3 factors: 1) feature pre-training; 2) structured prediction, and 3) least-squares loss.

**R6.** *On the term "structure".* We agree with R6 that the term "structure" can be confused with existing methods. We
will differentiate such literature and TASML, and clarify that the term denotes the use of structured prediction.

• *Access training tasks at test time.* We agree that non-parametric methods could be challenging for low-resources
settings (e.g. mobile devices). However, in settings such as AutoML services in data centers, access to some past data
is common, and trading-off model adaptation time to achieve better performance is a valid use case (e.g. AutoGluon
by Erikson et. al 2020). Further, top-$M$ filtering (see line 180) already limits memory usage, and we plan to further
mitigate the requirement in the future, e.g. by storing only salient tasks (e.g. Sparse GP by Seeger et al. 2003).
• *Clarify conditional meta-learning* Conditional meta-learning refers to methods that condition initial model parameters
on target tasks, followed by gradient-based model adaptation. Methods such as MAML is unconditional as it learns a
shared initial parameters for all tasks.
• *Relation with works [1,2,3,4,5] from R6.* We note that [2,3] were discussed in the paper (lines 86-97, 188). [1]
assumes knowledge about task hierarchy and not directly comparable to TASML. We will include [4, 5] as conditional
methods. However, they do not belong to our paper's focus of gradient-based meta-learning methods. We note that
TASML achieves better results against [2, 4]'s reported results (the rest didn't perform the same benchmark).

[Meta-Review · NeurIPS 2020]

The reviewers agreed that this paper brings an important and relevant contribution to the NeurIPS community, and presents comprehensive experiments to validate the proposed approach. The authors are strongly encouraged to revise the submitted paper according to the feedback in the reviews, including a discussion of multi-task learning, adding the requested clarifications, and fixing typos.